# Development of Hydrophobic, Anticorrosive, Nanocomposite Polymeric Coatings from Canola Oil: A Sustainable Resource

**DOI:** 10.3390/polym12122886

**Published:** 2020-12-01

**Authors:** Manawwer Alam, Naser M. Alandis, Naushad Ahmad, Fahmina Zafar, Aslam Khan, Mohammad Asif Alam

**Affiliations:** 1Department of Chemistry, College of Science, King Saud University, P.O. Box 2455, Riyadh 11451, Saudi Arabia; nandis@ksu.edu.sa (N.M.A.); anaushad@ksu.edu.sa (N.A.); 2Materials Research Laboratory, Department of Chemistry, Jamia Millia Islamia, New Delhi 110025, India; fahmzafar@gmail.com; 3King Abdullah Institute for Nanotechnology, King Saud University, P.O. Box 2455, Riyadh 11451, Saudi Arabia; aslamkhan@ksu.edu.sa; 4Center of Excellence in Engineering Materials, King Saud University, P.O. Box 800, Riyadh 11421, Saudi Arabia; moalam@ksu.edu.sa

**Keywords:** Canola oil, urethane, oxalic acid, fumed silica, nanocomposite

## Abstract

A novel hydrophobic Canola oil-based nanocomposite anticorrosive coating material with different contents of fumes silica (FS) was successfully synthesized via an in situ method. Firstly, a Canola oil-based hydroxyl terminated poly (oxalate-amide) was prepared by a two-step process of amidation and condensation. Secondly, the dispersion of fumed silica (1 to 3 wt.%) in hydroxyl terminated poly (oxalate-amide) was carried out, followed by reaction with toluene-2,4- diisocyanate (TDI) in order to form poly (urethane-oxalate-amide)/fumed silica nanocomposite. The structure and properties of nanocomposite were analyzed by FTIR, NMR (^1^H/^13^C), TGA/DTA, DSC, contact angle, and SEM. The physico-mechanical and electrochemical tests were performed in order to check the performance of nanocomposite coating. The results reveal that FS is homogenously dispersed in poly (urethane-oxalate-amide) matrix with a loading amount of less than 3 wt.%. The performance of nanocomposite coating improved when compared to virgin polymer. The synthesized nanocomposite coating can be used in the field of hydrophobic anticorrosive coatings.

## 1. Introduction

Corrosion is metallic cancer that cannot be completely eradicated, but can be overcome by the use of different methods and techniques. The use of polymeric coatings is one of the most important techniques for overcoming corrosion. Generally, coatings are petro-based. The use of renewable material for the synthesis of environmentally friendly low molecular weight polymers has attracted interests among scientists and technologists in the field of coatings [1]. This is due to fast the depletion of fossil fuels and day-by-day increase in their prices. Vegetable seed oils (VS) are one of the most important renewable materials amongst other bio-derivatives for the development of paints and coatings, along with other applications, like biodiesel, adhesives, inks, plasticizers, cosmetics, amphiphilic copolymers, biomedical, and others [2]. Biodegradability, non-toxicity, sustainability, multi-functionality, abundant availability, low cost, and their ease of transformation into value added low molecular polymers has led to versatile applications of triglyceride based VS (e.g. soybean, jatropha, linseed, sunflower, palm, castor, nahar, and canola oils) [2,3]. They are used as precursors in the production of polyols, epoxies, alkyds, polyurethanes, polyestermides, and polyetheramides for anticorrosive and anti-bacterial coatings applications [1,2,3,4]. The incorporation of nanofillers (metal or metal oxides, CNTs, Graphine, nanoclay, and fumed silica) enhances the physico-mechanical, electrical, antibacterial, and anticorrosive properties of VS based polymers [5,6,7,8,9,10,11]. Amongst these, fumed silica nano-fillers, due to their controlled size, large surface area, uniform structure, and remarkable moisture-barrier properties, provide improved strength, stiffness, and anticorrosive properties for advance applications [12,13,14]. Deewan et al. synthesized VS based polyurethane/fumed silica hybrid nanocomposite coatings for antibacterial and anticorrosive application [15]. M Alam et al. [16,17] used different VS based polyurethane nanocomposite with the dispersion of fumes silica (FS) and found that the incorporation of FS improved the properties of the respective virgin resin (Table 1).

The present work reports the synthesis of Canola (CA) oil poly (urethane-oxalate-amide)/FS nanocomposite coating material. CA oil is selected for the present work, due to its fatty acid composition: oleic (C18:1) 57%, linoleic (C18:2) 21.5%, and linolenic (C18:3) 10% unsaturated fatty acids. CA oil has been transformed to polyols, polyurethanes, water reducible alkyds, epoxies, and others [18,19]. 

FTIR, NMR (^1^H/^13^C), TGA/DTA, DSC, contact angle, XRD, and SEM have been used in order to characterize the nanocomposite. The physico-mechanical (scratch hardness, impact resistance, and bend test) and electrochemical tests was performed in order to check the performance of nanocomposite coating.

## 2. Materials and Methods

Canola (CA) oil (Afia International Company, Jeddah, Saudi Arabia), Oxalic acid (OA) (Fluka AG. Chemische Fabrik CH-9470 Buchs, Switzerland), Diethanolamine (WinLab, city, UK), toluene-2,4-diisocyanate (TDI) (Acros Organic, New Jersey, USA), fumed silica (FS) (Sigma Aldrich, Louis, USA), sodium metal, methanol, and toluene (BDH Chemicals Ltd., Poole, UK) are used as such without further purification.

CADFA, CAPOA, CAPUOA, and CAPUOA/FS were analyzed with a FTIR spectrophotometer while using KBr window and operated through spectrum 10 software, resolution 4 cm^−1^ wavelength 4000–400 cm^−1^ (Spectrum 100, Perkin Elmer Cetus Instrument, Norwalk, CT, USA). ^1^H NMR and ^13^C-NMR spectra were performed (JEOL DPX400MHz, Japan) while using deuterated Chloroform (CDCl_3_) as solvent and tetramethylsilane (TMS) as internal standard. Thermal analysis of CADFA, CAPOA, CAPUOA, and CAPUOA/FS was measured by TGA and DSC (Mettler Toledo AG, Analytical CH-8603, Schwerzenbach, Switzerland): TGA temperature range 25 °C to 600 °C and DSC temperature range of 80 °C to 90 °C/60 °C to 350 °C while using nitrogen atmosphere and heating rate 10 °C/min. The morphology of PUOEA/FS nanocomposite coating was studied by a field emission scanning electron microscopy (FE SEM, JSM 7600F, JEOL, Japan) that was operated at 5/15 KV and energy dispersive X- ray spectroscopy (EDX, Oxford, UK). Physico-chemical studies of CADFA, CAPOA, CAPUOA, and CAPUOA/FS were assessed by acid (ASTM D555–61) [22,23] and refractive index (ASTM D1218). The coating properties of nanocomposite were assessed by scratch hardness (BS 3900), crosshatch (ASTM D3359-02), pencil hardness test (ASTM D3363-05), impact test (IS 101 part 5 s−1,1988), flexibility/bending test (ASTM D3281-84), gloss (by Gloss meter, Model: KSJ MG6-F1, KSJ Photoelectrical Instruments Co., Ltd. Quanzhou, China), and thickness (ASTM D 1186-B). The hydrophobicity was measured by contact angle measurement by a CAM 200 Attention goniometer, while using 0.5 µL volume of distilled water drop on the surface of coated panel. CAPUOA and CAPUOA/FS nanocomposite coated carbon steel (CS) specimens were attended as working electrode; an exposed surface area 1.0 cm^2^ was fixed by PortHoles electrochemical sample mask, platinum electrode as counter electrode, and 3 M KCl filled silver electrode as reference electrode with Auto lab potentiostat/galvanostat, PGSTAT204-FRA32, with NOVA 2.1 software (MetrohomAutolab B.V. Kanaalweg 29-G, 3526 KM, Utrecht, Switzerland).

### 2.1. Synthesis of Canola Diol Fatty Amide (CADFA)

The procedure for the synthesis of CADFA was followed according to our previously published paper [22]. CADFA was prepared by a reaction of CA oil with diethanolamine in the presence of base catalyst. Thin layer chromatography (TLC) and FTIR confirmed the reaction.

### 2.2. Synthesis of CA Oil Based Poly (Oxalate-Amide) (CAPOA) (Step 1)

CADFA (2.0 mol) and OA (1.8 mol) were used in order to prepare CAPOA, as mentioned in a previously reported work [17]. The synthesis was carried out at a reaction temperature 120 °C and the reaction was monitored by acid value (28) determination along with FTIR.

### 2.3. Synthesis of CA Oil Based Poly(Urethane- Oxalate-Amide) (CAPUOA) (Step 2)

10 gm CAPOA was dissolved in toluene in a four necked flask adding 25 wt.%, 30 wt.%, and 35 wt.% TDI and placed over a magnetic stirrer (250 rpm) with a hot plate at a reaction temperature of 125 °C to carry out the synthesis reaction. The flask was fitted with a cold water condenser, thermometer, and nitrogen inlet. The progress of the reaction was monitored by hydroxyl value determination and TLC at regular intervals of time. The excess solvent was removed by a vacuum evaporator to recover CAPUOA. The samples were designated as CAPUOA25, CAPUOA30, and CAPUOA35; the last numeral indicates the % loading of TDI (Scheme 1).

### 2.4. Preparation of Poly(Urethane-Oxalate-Amide)/Fumed Silica Nanocomposite(Step 3)

CAPUOA30 was selected for the preparation of CAPUOA/FS nanocomposite on the basis of its film forming ability and mechanical performance [17]. FS was loaded into the polymer before urethane reaction, as in Step 2. 20 g of CAPOA was dissolved in 50 mL toluene and FS (1 wt.%, 2 wt.%, and 3 wt.%) was added slowly at room temperature (25 ± 5 °C) with continuous stirring. The components were stirred for 30 min. and then TDI (30 wt.%) was added drop wise. The flask was fitted with cold water condenser, thermometer, and nitrogen inlet. The reaction mixture was heated at 120 ± 5 °C and the progress of reaction was monitored by hydroxyl value determination and performing TLC at regular intervals. The obtained nanocomposites were designated as CAPUOA/FS1, CAPUOA/FS2, and CAPUOA/FS3, respectively.

### 2.5. Preparation of CAPUOA/FS Nanocomposite Coating

Carbon steel (CS) strips were polished with different grades of silicon carbide papers, washed with double distilled water, degreased with methanol and acetone, and dried at room temperature, before the coating preparation. CAPUOA/FS1, CAPUOA/FS2, and CAPUOA/FS3 resins were dissolved in toluene, 40% (*w*/*v*). The prepared solution was brushed on commercially available CS strips of standard size (70 mm × 25 mm × 1 mm) and composition (in weight %: 2.87% C and 97.13% Fe). The coated stripes were dried at room temperature for 15 days for complete curing. After complete curing, the coated panels were subjected to physico-mechanical and corrosion tests.

## 3. Results and Discussion

The scheme shows chemical reactions for the synthesis of CADFA, CAPOA, CAPUO, and CAPUOA/FS. It is clear from the reaction that CAPOA resin was synthesized in two steps—amidation, followed by the esterification. CAPUOA resin was synthesized in three steps—amidation, esterification, followed by urethanation. Similarly, nanocomposite (CAPUOA/FS) preparation involved amidation, esterification, and dispersion of FS, followed by urethanation. The dispersion of FS was safely carried out only up to 3 wt.% in CAPUOA30 matrix; beyond that, agglomeration was observed. In brief, the nanocomposite preparation was started from CA oil, which was transformed into CADFA that, on further reaction with OA and TDI, give CAPOA and CAPUOA, respectively. The dispersion of FS in CAPOA, followed by the reaction with TDI, formed CAPUOA/FS nanocomposite.

### 3.1. Solubility and Physico-Chemical Properties

The CAPUOA/FS nanocomposite shows a solubility in toulene, xylene, di ethyl ether, DMF, DMSO, ethyl methyl ketone, acetone, ethylacetone, THF, chloroform, carbon tetra chloride, dicholoromethane, benzene, pyridine, chloroform, 1,4 dioxane, and benzyl alcohol. While it is sparingly soluble in ethanol, methanol, formamide, *n*-butanol, *n*-propanol, and insoluble in *n*-hexane, water, and acetonitrile. The solubility of nanocomposite is due to its functional groups and long alky chain [23]. The sharp decrease in acid value was observed from 28 in CAPOA to 15 in CAPUOA; whereas, a slight decrease in acid value occurred from CAPUOA to CAPUOA/FS (CAPUOA/FS1, 13.90; CAPUOA/FS2, 13.50; and, CAPUOA/FS3, 13.28). The refractive index of oil was 1.463, and this increased in the order of CADFA (1.4905), CAPOA (1.5051), CAPUOA (1.5070), CAPUOA/FS1 (1.5112), CAPUOA/FS2v(1.5118), and CAPUOA/FS3v(1.5121). These results can be correlated to the formation of CAPOA from CADFA, followed by the formation of polyurethane and nanocomposites.

### 3.2. Spectral Analysis

Figure 1 provides the FTIR of CA oil, CADFA, CAPOA, CAPUOA, and CAPUOA/FS2. The FTIR (cm^−1^) spectra of CA oil shows the characteristic peaks of triglyceride. These peaks are 3007.34 (–HC=CH–); 2926.70 (CH_2_, asymmetrical), 2855.31 (CH_2_, symmetrical); 1746.24 (>C=O, ester); 1654.81 (–HC=CH–); 1461.93 (vibration of deformation–CH–); 1377.43 (deformation vibration of methylene group); 1238.27, 1163.95(–C(=O)–O–C–, ester), 1119.75 (–CO–CC–CC), and 722.97(–CH–bending). The FTIR spectrum of CADFA shows all of the characteristic peaks of alkyl fatty acid chain, along with appearance of additional bands and disappearance of some bands. The additional bands appeared at 3352.91 (–OH, hydrogen bonded) and 1060.84 (broad band of asymmetrical stretching CO in C–OH) in CADFA, ascribed to hydroxyl groups. The same band appears in the spectrum of CAPOA with lower intensity (non-hydrogen bounded), due to chemical reaction between –COOH group of OA and –OH group of CADFA. The –OH band was further suppressed and new sharp band appeared in the spectra of CAPUOA and CAPUOA/FS2 that indicates a chemical reaction occurred between the residual –OH group of CADFA and NCO group of TDI. The absorption bands at 2854–2855 cm^−1^ and 2924–2925 cm^−1^ are due to CH_2_ asymmetric and symmetric stretching. The peak at 3006–3007 cm^−1^ corresponds to the presence of unsaturation in the long alkyl chain.

The presence of the peak at 1738.7 cm^−1^ is attributed to the ester >C=O in case of CAPOA. In the spectra of CAPUOA and CAPUOA/FS2, the >C=O peak is observed at 1730 cm^−1^ to 1733 cm^−1^, respectively, which correlated to urethane in the >C=O peak. The appearance of peak at 1618.53 (CO, amide) and 1455.47 (CN, amide), and the disappearance of ester bands is related to amidation of CA oil to form CADFA [5,17].

### 3.3. NMR Spectra

The structure of the polymeric resins was further confirmed by comparing the NMR spectra of CADFA, CAPOA, and CAPUOA. Figure 2 and Figure 3 show the ^1^H-NMR and ^13^C–NMR spectra of CADFA, CAPOA, and CAPUOA. In the CADFA spectra (Figure 2), the disappearance of ester carbonyl peak (CH/CH_2_-OCO) and appearance of some additional peaks at 5.2 ppm (OH), 3.6–3.7 ppm (–CH_2_OH), 3.4–3.5 ppm (–CH_2_–N–), along with all of the characteristic peaks of long alkyl unsaturated chain, was observed. The characteristic peaks of unsaturated chains are found at 2.3 ppm (CH_2_–CO), 1.5 ppm (CH_2_–CH_2_–CO), 2.0 ppm (CH_2_–CH=CH), 5.5 ppm (CH=CH), 1.9 ppm (CH=CH–CH_2_), 2.9 ppm (=CH–CH_2_–CH=), 1.83 ppm (chain CH_2_), and 0.86 ppm (–CH_3_). These observations can be related to the amidation of CA oil [18]. Because of the chemical reaction with OA, the suppression of OH (5.2 ppm) and –CH_2_OH (3.6–3.7 ppm) peaks and the appearance of peak at 4.14 ppm (CH_2_–OCO) and 7.96 ppm (–COOH) is observed (Figure 2). The presence of characteristic peaks of urethane, such as peak at 7.546 ppm (NH of urethane), 4.0–3.98 (CH_2_ attached to urethane carbonyl), and 2.22 ppm (CH_3_ of TDI) confirm the urethanation reaction. The ^13^C NMR spectrum of CAPUOA reveals the peak for urethane carbonyl (153.5 ppm), aromatic carbons (128.5 ppm, 130.2 ppm, 135.2 ppm, 137.5 ppm, and 136.3 ppm), and CH_3_ of TDI at 16.4 ppm correlated to the formation CAPUOA via urethanation (Figure 3) [23].

### 3.4. XRD Analysis

Figure 4a shows the XRD pattern of CAPUOA and CAPUOA/FS2. It reveals broad peak 2 theta at 20° that is correlated to the amorphous behavior of both the materials. This behavior did not affect the dispersion of FS [24,25].

### 3.5. Surface Wettability

The contact angle is measured in order to elucidate the surface wettability characteristic of the coatings. Figure 4b shows contact angle of CAPUOA and CAPUOA/FS2. It reveals the hydrophobic behavior (Contact angle >90°) of CAPUOA and CAPUOA/FS2 coatings. The hydrophobic behavior increases with the loading of FS, as in the case of CAPUOA/FS2 coatings (Contact angle = 108°). The increasing surface hydrophobicity from CAPUOA to CAPUOA/FS2 can be related to the increase in the roughness on the surface, due to the homogenous dispersion of FS nanoparticle within the matrix of CAPUOA [26]. Additionally, the FS particles fill voids in the polymer matrix, which increases the adhesion, roughness, barrier, and anticorrosive properties of coatings [27,28].

### 3.6. SEM and EDX Analysis

Figure 5 illustrates the FE-SEM morphological characterization of CAPUOA/FS2 film and distribution of FS in the polymeric matrix. Figure 5a,b present the FE-SEM images for the prepared CAPUOA/FS2 specimen at two different resolutions (×13,000, and ×40,000). The bright (white) spot in the image represents silica oxide particles in CAPUOA/FS2 nanocomposite. It is clear from the image that the homogenous distribution of fumed silica in the matrix enhances the properties of CAPUOA/FS2. The compactness and enhancement in electrostatic interactions at the coating metal surface increases with increased FS loading (occurring as aggregates), which leads to a deterioration of the coatings performance (CAPUOA/FS3) [27]. EDX and elemental mapping analysis were used to confirm the dispersion of silica oxide in polymeric matrix. The weight % of silica on the coating surface was found to be 1.97% in EDX analysis (Figure 6). This % of silica is consistent with the amount of FS that is added to the nanocomposite. Figure 6 reveals the specific location of carbon (ice blue), oxygen (blue), nitrogen (yellow), along with silica (red), which confirms the presence of silica oxide nanoparticles within the matrix.

### 3.7. Physico-Mechanical Characterization

It was observed that CAPOA coating was cured at high temperature (180 °C for 30 min). The CAPUOA, CAPUOA/FS coatings were obtained/cured at room temperature. The curing process of the coating involved physical and chemical processes. The solvent evaporation (first step) is a physical process, while the reaction of free NCO with atmospheric moisture (polyurethane secondary reactions) and auto-oxidation at alkyl chain are chemical processes [23]. The drying time decreased with the increased percentage of FS loading. This can be related to the uniform dispersion of filler and adequate electrostatic interactions between polar functionalities of FS and polymer matrix, which lead to higher cross linking within CAPUOA/FS coatings materials when compared to CAPUOA [17]. Because of this, the coating showed good physico-mechanical performance up to certain limits.

The thickness of the coating was found to be in the range of 200 to 235. Table 2 shows the physico-mechanical parameters of CAPUOA, CAPUOA/FS1, CAPUOA/FS2, and CAPUOA/FS3 coatings. It reveals that the scratch hardness, pencil hardness, and gloss (at 60°) increase with the increased % of FS. All of the coatings passed the cross-hatch test (100%). Where the CAPUOA/FS1 and CAPUOA/FS2 coatings passed the bending test (1/8 inch) and impact tests (100 lb/inch), the CAPUOA/FS3 coating failed the latter two tests that can be due to increased hardness, denseness, and excessive cross linking with 3 wt.% FS inclusion. The performance of CAPUOA/FS1 and CAPUOA/FS2 are related to the interaction of polar groups of coating materials with CS surfaces that enhance the adhesion at the coating and metal interface, and the crosslinking density leads to infusible thermoset. The particle–particle interactions and compactness in the aggregates of FS increases at higher loading, as in the case of CAPUOA/FS3, which increases rigidity and stiffness, leading to a decrease in the coating performance [18].

### 3.8. Thermal Analysis

Figure 7 shows the TGA and DTG thermograms of CAPUOA, CAPUOA/FS1, CAPUOA/FS2, and CAPUOA/FS3. It reveals materials exhibited three step degradation behavior. Generally, polyurethane follows a two-step degradation process. The thermal behavior of polyurethanes is affected by many factors, like nature of raw materials, ratio and type of soft/hard segment, crosslink density, and synthesis route [6,7]. The initial weight loss (2–5%) below 200 °C can be related to the removal of solvents/water (chemically and physically trapped). The first step of degradation initiated at around 252 °C (maximum degradation rate, T_max_ at 255 °C), second degradation step at around 300 (T_max_ = 350 °C), and third step of degradation at 390 °C (T_max_ = 450 °C). The first step is related to the degradation of hard segment, while the second and third steps correlated to soft segments degradation. The same degradation behavior was also observed in nanocomposites. The thermal behavior of the polymers is not noticeably affected by the dispersion of FS.

Figure 8a shows the DSC thermogram of CA oil and CADFA. The DSC thermogram of CA oil shows large endotherm from −50 to 0 °C (centered at −20 °C) and the CADFA thermogram endotherm is observed from 50 to 64 °C (centered at 55 °C). Figure 8b shows the DSC thermogram of CAPUOA, CAPUOA/FS1, CAPUOA/FS2, and CAPUOA/FS3. It reveals a broad endotherm centered at 228 °C (starts from 175 to 275 °C), 238 °C (starts from 188 to 285 °C), 242 °C (starts from 192 to 290 °C), and 244 °C (starts from 190 to 296 °C), respectively. These endotherms are related to their melting temperature. It is observed from figure that melting temperature increases with the inclusion of FS.

### 3.9. Corrosion Study

#### 3.9.1. Potentiodynamic Polarization

The Tafel polarization curves (Figure 9) for bare carbon steel (CS) and CAPUOA/FS2 nanocomposite were conducted in the 3.5% *w*/*v* NaCl corrosive solution. Information regarding corrosion rate (CR), corrosion potential (E_corr_), and corrosion current can be obtained by the Tafel extrapolation method. Table 3 illustrates the corrosion parameters for bare CS and CAPUOA/FS2 nanocomposite coating in 3.5% NaCl solution. Note that the CAPUOA/FS2 nanocomposite coating possesses a higher corrosion potential than bare CS, which indicates that the CAPUOA/FS2 nanocomposite coating shows good corrosion protection for CS. However, it is clear from Table 4 that the corrosion potential of CAPUOA/FS2 nanocomposite coating significantly increases, and the corrosion current reduces. These results indicate that CAPUOA/FS2 nanocomposite can act as a protective layer on CS and improve the anti-corrosion performance. The Tafel polarization curves exhibit that the CAPUOA/FS2 nanocomposite coating causes a positive displacement in the corrosion potential, relative to the value of the bare CS. This shift in the corrosion potential of CAPUOA/FS2 nanocomposite coatings towards the bare CS during the immersion time (three to 18 days) increases. Tafel measurements clearly show that a substantial reduction in corrosion rate occurs for the CAPUOA/FS2 nanocomposite coated sample with respect to bare CS. This reduction may be due to the inhibitory effect of corrosion ions. However, the CAPUOA/FS2 nanocomposite coating can act as corrosion barrier for CS and reduce the rate of corrosion during the immersion in corrosive solution [29,30,31].

#### 3.9.2. Electrochemical Impedance Spectroscopy (EIS)

EIS is a measurement technique, which allows for a wide variety of coating evaluations. The corrosion performance of bare CS and CAPUOA/FS2 nanocomposite coated on CS was also investigated by EIS during various (3, 6, 9, 12, 15, 18 days) exposure times in 3.5% *w*/*v* NaCl solution. EIS presents the classical behavior that is often described in the literature [32,33,34]. For bare CS and CAPUOA/FS2 nanocomposite throughout (18 days) various immersion times, only one capacitive loop is detected, and the loop size decreases with increasing immersion time, as shown in Figure 9. Such an electrochemical behavior can be described in terms of a simple equivalent circuit depicted in Figure 8, which consists of the parallel combination of a dielectric capacitor/coating capacitance (C_c_) and resistive components, such as R_ct_ and R_s_. Table 4 shows changes in the magnitudes of these parameters, as a function of their exposure times (3, 6, 9, 12, 15, 18 days). Cc increases with the immersion time, due to water uptake, whereas R_ct_ decreases with time. The R_ct_ values obtained for PUOEA/FS2 nanocomposite coating are relatively high with respect to the observed bare CS. The maximum amplitude of the impedance signal was observed after three day exposure with increasing immersion time, as depicted in Nyquist plots (Figure 9). This indicates that the corrosion process has not been initiated on substrate with CAPUOA/FS2 nanocomposite coatings still preventing the metal to come in direct contact with the aqueous saline environment. Table 4 provides the changes in the values of Cc and R_ct_ of CAPUOA/FS2 nanocomposite coating versus exposure time. The presence of fumed silica in CAPUOA matrix effectively increases the length of the diffusion pathways for oxygen and water and decreases the permeability of the coating, which leads to higher corrosion resistivity of CAPUOA/FS2 coating. This increase is related to the lowering of charge transfer rate between the metal and the solution. The charge transfer reactions are known to take place at the metal/nanocomposite interfaces. Consequently, the high R_ct_ values of CAPUOA/FS2 nanocomposite can be explained by the effective barrier behavior of the CAPUOA/FS2 nanocomposite coating. However, barrier properties of coating decreased with increased immersion time [28,29,30]. From the EIS measurements and potentiodynamic results, it can be concluded that the immersion time has a significant effect on the CAPUOA/FS2 nanocomposite coating. In a short immersion time, perfect coatings often demonstrate a good barrier performance resulting in a large impedance (high frequency) of metal/CAPUOA/FS2 system, while small impedance (low frequency) shows that the metal substrate has been corroded [34].

## 4. Conclusions

A vegetable oil is transformed into nanocomposite by a simple method to combat corrosion. FTIR, SEM, and TGA were used to characterize the structure, morphology and thermal stability of the nanocomposite. The coatings demonstrated a remarkable improvement on the basis of their physico-mechanical properties, such as gloss, scratch hardness, pencil hardness, impact resistance, bend test, and cross hatch with the inclusion of FS. The CAPUOA/FS2 nanocomposite showed best physico-mechanical properties (gloss: 100 °C, scratch hardness: 3 kg, impact (100 lb/inch passes), bend (1/8 inch passes), amongst themselves. The corrosion performance of the CAPUOA/FS coatings was carried out by Tafel and EIS analysis under 3.5 wt.% NaCl. It was found that CAPUOA/FS2 nanocomposite coatings exhibited the highest corrosion rate 8.831 × 10^−5^ mm/y. A uniform dispersion of FS resulted in the formation of strong secondary barrier, which helped in the enhancement of anticorrosive activity of CAPUOA coatings. The studies revealed that the CAPUOA/FS nanocomposite coatings have potential scope for their application in the field of anticorrosive coatings.

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
