# Peer review of "Development of Hydrophobic, Anticorrosive, Nanocomposite Polymeric Coatings from Canola Oil: A Sustainable Resource"

_polymers, 2020, doi:10.3390/polym12122886_

Round 1
Reviewer 1 Report
This manuscript presents a preparation of nanocomposite anticorrosive coating from a natural product, i.e. canola oil. Such coating is biodegradable, non-toxic, sustainable, and inexpensive.
The manuscript presents compressive characterization; however, the presentation of the result should be improved intensively.
- Scheme 1, step 3 is not clear, should be rebuilt. Do the beads denote FS?
- Figure 1, it would be better if the denoted wavenumbers are replaced by the groups.
- The resolution of Figures 2 & 3 should be improved. It’s hard to recognize the chemical shift. Also, the name of the x-axial (chemical shift, ppm) should be provided.
- Table 3 & 4, Figure 9 are messy. What is the data for CAPUOA/FS2. More information can be provided in the caption.
Author Response
Reviewer 1
- Scheme 1, step 3 is not clear, should be rebuilt. Do the beads denote FS?
Response: Scheme 1, step 3 has been improved. Yes, beads denote FS
- Figure 1, it would be better if the denoted wavenumbers are replaced by the groups.
Response: Figure 1 modified according to reviewer suggestion
- The resolution of Figures 2 & 3 should be improved. It’s hard to recognize the chemical shift. Also, the name of the x-axial (chemical shift, ppm) should be provided.
Response: The resolution of Figures 2 & 3 has been improved. The name of the x-axial (chemical shift, ppm) has been provided.
- Table 3 & 4, Figure 9 are messy. What is the data for CAPUOA/FS2. More information can be provided in the caption.
Response: Table 3 & 4, Figure 9 has been improved. The data for CAPUOA/FS2 has been provided in the caption. I screened all nanocomposite of CAPUOA/FS based of Physico-mechanical and drying properties therefore I have done corrosion test only CAPUOA/FS2 coated MS.

Reviewer 2 Report
The work deals with hydrophobic canola oil based nanocomposite anticorrosive coating material with different contents of fumes silica. Samples were analysed by different techniques both morphological and physico-mechanical. Moreover, electrochemical test was done.
The topic is interesting and quite new, but the paper needs a lot of work. Moreover, English must be improved.
Material and method:
It’s difficult to understand the different preparation step sequence.
You prepared CAPUOA25, CAPUOA30 and CAPUOA35, but finally you never tested it. In the experimental is written: “It was found that CAPUOA/FS2 showed best coating properties”. This isn’t the correct position for this sentence. I deduced that this is the reason why, for some test, you tested only CAPUOA/F2. Moreover, in the experimental section, is necessary to add information about analysis. For example: FTIR (operating range), TGA (maximum temperature), Contact angle (liquid used).
Line 101. What do you mean for “desired hydroxyl value”
Results:
In this section, you show the results of FTIR, XRay, contact angle SEM and EDX of only CAPUOA and CAPUOA/FS2 but you made Physico-mechanical characterization of CAPUOA/FS1, CAPUOA/FS2, and CAPUOA/FS3. What is the reason for this?
Line 224: you wrote: “All the coatings pass cross hatch, bending test (1/8 inch) and impact tests”. Looking in table 2,
it’s written that CAPUOA/FS3 fail bending and impact test.
Author Response
Reviewer 2
Material and method:
It’s difficult to understand the different preparation step sequence.
Response:It has been improved.
You prepared CAPUOA25, CAPUOA30 and CAPUOA35, but finally you never tested it.
Response:We have tested CAPUOA30 (denoted as CAPUOA to simplify) that selected on the basis of its good film forming ability as compared to other compositions (CAPUOA25 and CAPUOA35). The structure of CAPUOA by using FTIR and NMR to confirm the formation of urethane linkages along with ester linkages (Figure1). Other characterization (Contact angle, XRD, mechanical, TGA/DTG and DSC of CAPUOA has also been mentioned in the manuscript.
In the experimental is written: “It was found that CAPUOA/FS2 showed best coating properties”. This isn’t the correct position for this sentence.
Response:Yes, It is not a correct position. It has been deleted from the section.
I deduced that this is the reason why, for some test, you tested only CAPUOA/F2. Moreover, in the experimental section, is necessary to add information about analysis. For example: FTIR (operating range), TGA (maximum temperature), Contact angle (liquid used).
Response: The aim of the work is to develop coating for corrosion protection. On the basis of Physico-mechanical results. We have screen out the system CAPUOA/FS2 with best performance. So, we have used only this system for other characterization. We included the FTIR(operating range),TGA(Temperature range), Contact angle(liquid used) and other characterization conditions in revised manuscript.
Line 101. What do you mean for “desired hydroxyl value”
Response: The desired hydroxyl value means the reaction is stop at the required value of hydroxyl group for their further reaction with TDI to form polyurethane. In the present case the calculated HV for the reaction is 100 ± 0.5 mg KOH/g. We have already tested the reaction at higher and lower HV and found best results at HV (100 ± 0.5 mg KOH/g).
Results:
In this section, you show the results of FTIR, XRay, contact angle SEM and EDX of only CAPUOA and CAPUOA/FS2 but you made Physico-mechanical characterization of CAPUOA/FS1, CAPUOA/FS2, and CAPUOA/FS3. What is the reason for this?
Response:The results of FTIR were just used to confirm the reaction. So we have shown FTIR of CA oil, CADFA, CAPOA, CAPUOA and CAPUOA/FS2 (Figure 1). Since, the reaction started from the oil. So it can be easily understand the involvement of function group during the reaction. Physico-mechanical characterization of all the CAPUOA/FS1, CAPUOA/FS2, and CAPUOA/FS3 coatings were carried out to obtain best composition for mechanically strong system for corrosion resistance. On the basis of these results, we have screen out the system CAPUOA/FS2 with best performance. So, we have characterized XRD, contact angle of CAPUOA/FS2 and CAPUOA just for comparison and SEM, EDX only CAPUOA/F2.
Line 224: you wrote: “All the coatings pass cross hatch, bending test (1/8 inch) and impact tests”. Looking in table 2,it’s written that CAPUOA/FS3 fail bending and impact test.
Response:Thank you for the comment. We have been corrected in the manuscript as “All the coatings pass cross hatch test. Whereas CAPUOA/FS1 and CAPUOA/FS2 coatings pass bending test (1/8 inch) and impact tests. CAPUOA/FS3 coatings fail last these two tests that can be due to increase hardness, denseness and excessive cross linking with 3 % FS”.
Reviewer 3 Report
- Line 276, “We suggest that the metal/polymer interface and are released during immersion course of time.” As the immersion test is only 18 days, this suggests the coating quality is low.
- Line 292, “Table 4, Cc increases with the immersion time due to water uptake, whereas Rct decreases with time.” Same thing, this suggests low coating quality?
- Line 312, Figure 9.
First, the equivalent circuit doesn’t match the description of the text. For example, where is Rs? Also, it is questionable to use a CPE instead of using a double layer (double-layer capacitor in parallel with a charge- transfer reaction resistor).
Secondly, in figure 9(b), the reason of a significant change of OCP (Ecorr) between 6 days and 9 days is unclear.
Author Response
Reviewer 3
- Line 276, “We suggest that the metal/polymer interface and are released during immersion course of time.” As the immersion test is only 18 days, this suggests the coating quality is low.
- Line 292, “Table 4, Cc increases with the immersion time due to water uptake, whereas Rct decreases with time.” Same thing, this suggests low coating quality?
Response:This coating materials developed by vegetable oil, which is degradable, while time increases the coating performance will be inferior in continued dipped in an electrolyte for longer periods(18 days).
- Line 312, Figure 9.First, the equivalent circuit doesn’t match the description of the text. For example, where is Rs? Also, it is questionable to use a CPE instead of using a double layer (double-layer capacitor in parallel with a charge- transfer reaction resistor).
Response: Thankyou for point out the good comments, we corrected the circuit CPE is not used in place of double layer capacitance. we used CPE to adjust the circuit fitting.
The CPE is often used simply as a way to improve the fit of a model to impedance data, justified by a vague assertion that a distribution of time constants is present in the system under investigation. Ref. doi/10.1002/9781119363682.ch14
Secondly, in figure 9(b), the reason of a significant change of OCP (Ecorr) between 6 days and 9 days is unclear.
Response: we explained the reason here, during this period pores were developed in coating and electrolyte penetrates to the layer of coating.After 9 days penetration of coating stopped due to excess ions made a thick layer on the surface of pore and do not allow to enter ions on to the metal surface. Between 6 days and 9 days showed the significant change. After that slow changes. Ref.https://dx.doi.org/10.1021/acsomega.0c03333
Reviewer 4 Report
The summary is clear and describes the contribution of the study.
A state of the art is presented where the different studies carried out on the subject are discussed appropriately. Highlighting the area of opportunity of this study.
The methods are described correctly.
Line 125-133: Mention something about the characterization/validation techniques of the indicated products. While the following sections describe it, point out in this section with scientific support if possible.
Figure 2 and 3 can be improved (quality).
Linea 206-207: “It is clear from the image that homogenous distribution of fumed silica in the matrix that enhances the properties of CAPUOA/FS2”. Expand analysis with scientific support. Figure b, blurry. If possible, improves quality. Play with glitters and contrasts in the original micrograph.
Linea 219-222. “This can be related…….” Because? Expand discussion on this, with scientific support.
In general it is recommended to discuss results with scientific support, with the aim and improve the quality of the writing. Which is an interesting topic, backed by the different studies carried out. It ranges from synthesis to possible application of the material.
The conclusions are general and the following question should be answered (line 316-317): “The coating demonstrated a remarkable improvement in physico-mechanical properties” Why? based on the results and reled studies, conclude in this regard.
Author Response
Reviewer 4
Line 125-133: Mention something about the characterization/validation techniques of the indicated products. While the following sections describe it, point out in this section with scientific support if possible.
Response: Characterization techniques explained in the section of materials and methods.
Figure 2 and 3 can be improved (quality).
Response: Figure 2 and 3 has been improved.
Linea 206-207: “It is clear from the image that homogenous distribution of fumed silica in the matrix that enhances the properties of CAPUOA/FS2”. Expand analysis with scientific support. Figure b, blurry. If possible, improves quality. Play with glitters and contrasts in the original micrograph.
Response: Thank you very much for your valuable comments. As per the reviewers suggestion we have expanded the FE-SEM analysis in the revised manuscript. Moreover, we have improved the figure 5b contrast for better visibility in the revised manuscript and also provided the reference.
Linea 219-222. “This can be related…….” Because? Expand discussion on this, with scientific support.
Response: It has been modified as suggested by reviewer.
In general it is recommended to discuss results with scientific support, with the aim and improve the quality of the writing. Which is an interesting topic, backed by the different studies carried out. It ranges from synthesis to possible application of the material.
Response: We have been improved as suggested by reviewers.
The conclusions are general and the following question should be answered (line 316-317): “The coating demonstrated a remarkable improvement in physico-mechanical properties” Why? based on the results and reled studies, conclude in this regard.
Response: Conclusion has been improved as the reviewer’s suggestions.
Round 2
Reviewer 1 Report
The manuscript has been significantly improved and now can be published in Polymers.
Reviewer 2 Report
After the first revision the work was improved, but is necessary to add, for a correct sample comparion, corrosion test and SEM images of all the CAPUOA/FS.
The authors write: The aim of the work is to develop coating for corrosion protection, but made corrosion test only on CAPUOA/FS2. If is the most important factor, why did you do corrosion study only for CAPUOA/FS2? For a correct comparison is necessary to test all the samples.
Line 221: As the compactness and enhancement in electrostatic interaction at coating metal surface increases with the FS (occurs as aggregates) leading to deterioration of the coatings performance (CAPUOA/FS3). Why did you mention CAPUOA/FS3. Can you provide SEM picture of CAPUOA/FS1 and CAPUOA/FS3 in order to see the different distribution?
Line 319. Which is the difference between PUOEA/FS2 and CAPUOA/FS2?